# ORION: Decoupling and Alignment for Unified Autoregressive understanding and generation

**Taihang Hu[1], Mengting Chen[2*], Jinsong Lan[2], Xiaoyong Zhu[2], Kaifu Zhang[2]**
**Ming-Ming Cheng[2], Bo Zheng[2†], Yaxing Wang[1†]**

[1]VCIP, College of Computer Science, Nankai University    [2]Alibaba Group

{hutaihang00}@gmail.com, {cmm, yaxing}@nankai.edu.cn
{cmt271286, jinsonglan.ljs}@alibaba-inc.com

## Abstract

Unified multimodal Large Language Models (MLLMs) hold great promise for seamlessly integrating understanding and generation. However, monolithic autoregressive architectures, despite their elegance and conversational fluency, suffer from a fundamental semanticstructural conflict: optimizing for low-level reconstructability in generation leads to catastrophic forgetting of high-level semantic understanding. We present ORION, a unified framework that resolves this conflict through Decoupling and Alignment. A non-linear vision head decouples structural pressures from shared representations, while a novel Representation Consistency Loss explicitly aligns semantics during generation. Together with a curated progressive training recipe and high-quality multimodal data, our method enables balanced optimization of both capabilities. Built purely on a monolithic autoregressive backbone without task-specific separate parameters, ORION achieves performance on par with or exceeding recent state-of-the-art unified models that rely on more complex designs. These results validate monolithic autoregression as a simple, effective, and competitive path toward truly integrated multimodal intelligence.

## 1 Introduction

In recent years, multimodal artificial intelligence has advanced significantly along two parallel tracks. On one hand, Multimodal Large Language Models (MLLMs) have demonstrated powerful capabilities in complex cross-modal understanding tasks(Hurst et al., 2024; Liu et al., 2023; Bai et al., 2025; Chen et al., 2024b). On the other hand, diffusion models have achieved unprecedented realism and controllability in image generation(Ho et al., 2020; Rombach et al., 2022; Podell et al., 2023; Esser et al., 2024; Wu et al., 2025c; Zhang et al., 2023; Huang et al., 2024; Wu et al., 2025a). However, this functional separation of understanding and generation capabilities represents a critical bottleneck on the path toward general-purpose multimodal intelligence.

Fusing these two capabilities within a unified framework is the next frontier in the field. Recent explorations have primarily revolved around three architectural choices, as shown in Fig. 1. Cascaded architectures(Wu et al., 2025a; Lin et al., 2025a; Pan et al., 2025; Liu et al., 2025) serially connect understanding and generation models, essentially using the MLLM as a multimodal text encoder for the diffusion model, which makes it difficult to support native multi-round dialogue with interwoven text and images. Parallel architectures(Deng et al., 2025; Shi et al., 2024; Mo et al., 2025) decouple the parameters required for generation and understanding, but face challenges of high training costs and scalability. In contrast, a monolithic autoregressive architecture(Sun et al., 2023; Tong et al., 2024; Team, 2024), capable of processing and generating arbitrarily interleaved image-text sequences in an end-to-end manner, may be the most promising path forward. This approach treats images as another "language", predicting them through a unified autoregressive process, which provides the natural framework for fine-grained, controllable, and conversational content creation. Although prior works like Emu(Sun et al., 2023) and Chameleon(Team, 2024) have explored this direction, their performance has not yet reached a satisfactory level, suggesting that we still need

---

*Project leader. † Co-corresponding authors.

to explore more effective training strategies to unlock the full potential of monolithic architectures. Among the challenges, one of the most acute and prevalent is that when a powerful, pre-trained MLLM is generatively fine-tuned by adding a regression loss for visual tokens, the model's original understanding capabilities suffer from catastrophic forgetting.

We formalize this problem as a representational conflict between **Semantic Fidelity and Structural Reconstructability** within the model. Understanding tasks, driven by a cross-entropy loss, pursue high-level semantic fidelity, which requires the model's hidden representations to be highly separable in semantic space to facilitate classification. In contrast, generation tasks, driven by a Mean Squared Error (MSE) loss, pursue structural reconstructability, which demands that hidden representations contain sufficient low-level informationcan to precisely reconstruct their coordinates in a continuous embedding space. These two conflicting optimization objectives create a "tug-of-war" within the shared representation space. Our experiments provide clear evidence for this conflict. As shown in Fig. 3, naive generative fine-tuning causes severe semantic drift, with categorical predictions for visual tokens deviating from the plausible text semantics of the base model. Likewise, the ablation study (Tab. 4, first row) shows that simple full-parameter fine-tuning yields poor understanding and generation performance, underscoring the impact of the semanticstructural conflict.

To address this, we propose a systematic framework designed to reconcile this conflict. Methodologically, the core of this framework lies in **decoupling and alignment**. First, we decouple the direct pressure on the shared representations from these two tasks using a high-capacity non-linear vision head. This vision head is dedicated to translating the high-level semantic information from the LLM backbone into precise, low-level structural vectors. Second, we introduce a representation consistency supervision mechanism that explicitly aligns the conflicting objectives during optimization, ensuring the model does not lose its semantic understanding of visual content while learning structural reconstruction. To effectively implement this framework, we further construct a complete and efficient training regime. This includes: a curated three-stage progressive training strategy designed to introduce the generation task to preserve pre-trained semantic knowledge smoothly; a comprehensive collection and cleaning of high-quality, open-source datasets for both image understanding and generation, providing a solid foundation for learning complex cross-modal representations; and a series of engineering optimizations to ensure the efficient training of this complex system.

Evaluations on widely-used multimodal understanding and generation benchmarks(Yue et al., 2024; Liu et al., 2024c; Ghosh et al., 2023) show that our model, with a native autoregressive monolithic architecture and no separated parameters, achieves performance in both image understanding and generation that is comparable to recent unified models and surpasses previous monolithic architectures.

The main contributions of this paper are as follows:

- Conceptually, we are the first to identify and define the "semantic-structural representation conflict" in unified autoregressive MLLMs and argue that the monolithic autoregressive architecture is a key path toward supporting native multi-round image-text dialogue.

- Technically, we propose and validate a methodology of "decoupling and alignment", complemented by a curated, complete progressive training strategy and data support, forming a systematic solution, and will open-source all training code, data, and models to give back to the community

- Performatively, under a monolithic autoregressive architecture without separable parameters our model achieves combined understanding and generation performance comparable to that of recent works using other architectures, strongly validating the significant potential of this technical route.

## 2 RELATED WORK

### 2.1 UNIFYING GENERATION AND UNDERSTANDING

Recent efforts to unify visual generation and understanding have pursued three primary architectural paradigms (Fig. 1): **1)Cascaded Architectures.** This approach (Wu et al., 2025a; Lin et al., 2025a) cascades a Multimodal Large Language Model (MLLM) with a separate diffusion model, where the MLLM acts as a text encoder to guide image synthesis. While leveraging specialized models, this design cannot comprehend its own visual outputs, precluding multi-turn interaction. **2) Parallel Architectures.** These models (Deng et al., 2025; Shi et al., 2024; Mo et al., 2025; Lin et al., 2025b) use partially independent modules for different tasks within a unified framework. For instance, some

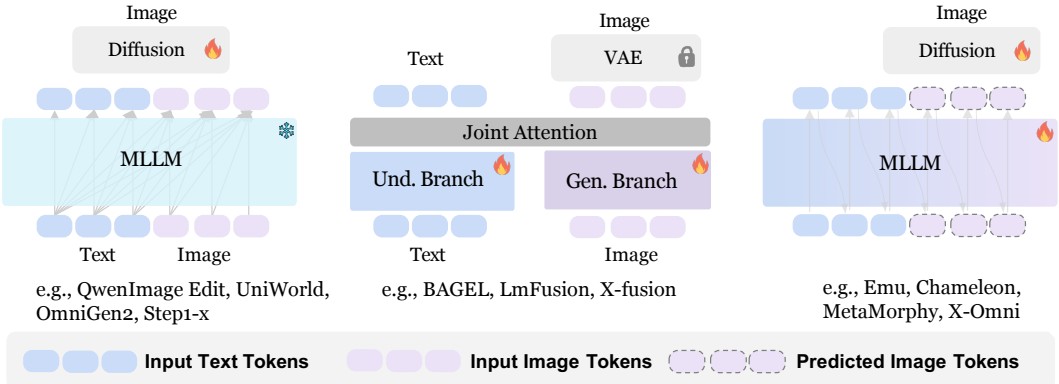

Figure 1: Three main architectural for recent unified models. (Left) Cascaded Architectures (Middle) Parallel Architectures. (Right)Monolithic Autoregressive Models

only share attention mechanisms between vision generation and understanding (Deng et al., 2025; Shi et al., 2024). This strategy typically results in large parameter counts and complex training pipelines. **3) Monolithic Autoregressive Architectures.** This paradigm (Sun et al., 2023; Team, 2024; Kou et al., 2024; Geng et al., 2025) processes interleaved text and image sequences with a single set of parameters under a unified autoregressive objective. Images are handled either by predicting continuous embeddings for a separate decoder (Sun et al., 2023) or by using discrete visual tokens for end-to-end training (Team, 2024). This unified design is elegant and naturally supports seamless multi-turn dialogue.

## 2.2 CONTINUOUS TOKENS VERSUS DISCRETE TOKENS

Early autoregressive models like LLamaGen(Sun et al., 2024) and Lumina-mGPT(Liu et al., 2024b) used a VQ-VAE(Van Den Oord et al., 2017) to discretize images, framing generation as a text-like prediction task with a unified cross-entropy loss. However, these VQ-VAE tokens prioritize low-level textures over high-level semantics, hindering the image understanding of models like Chameleon(Team, 2024). More recently, X-Omni(Geng et al., 2025) attempted to mitigate this by discretizing the outputs of a semantically-rich encoder (e.g., SigLIP(Zhai et al., 2023)) and using a diffusion decoder for image synthesis. While this improved multimodal comprehension, it still lagged behind methods using continuous semantic tokens. To maximally preserve multimodal understanding, our work leverages continuous image tokens.

## 2.3 PREDICTING IMAGE TOKENS AUTOREGRESSIVELY OR IN PARALLEL

MetaQuery(Pan et al., 2025) and Seed-X(Ge et al., 2024) simplify training by using a fixed set of query tokens to parallel regress image tokens. However, this creates a critical representational mismatch: the model generates from proxy tokens but must understand real image tokens. This inconsistency prevents the model from comprehending its own output, thus failing in coherent, multi-turn dialogues. In contrast, the purely autoregressive approach of serially predicting each visual token, while more challenging to train, ensures representational consistency, which is crucial for enabling seamless multimodal interaction.

# 3 METHODOLOGY

## 3.1 OVERVIEW OF MODEL ARCHITECTURE

The overall model architecture is depicted in Fig. 2, which is built upon a pretrained autoregressive MLLM. This model is capable of processing arbitrarily interleaved sequences of text and images. At the input stage, text is tokenized into a sequence of text tokens $T = \{t_1, t_2, ..., t_n\}$, while images are first encoded by a visual encoder into a series of continuous visual embedding, which are then mapped into a sequence of visual tokens $V = \{v_1, v_2, ..., v_m\}$. The core of the model is a Transformer decoder that autoregressively predicts the next token based on all preceding tokens, whether visual or textual. We denote the output from the final hidden layer of the Transformer at step $t$ as $h_t$. Within our unified framework, the model must handle two fundamental prediction tasks, supervised by the following two loss functions:

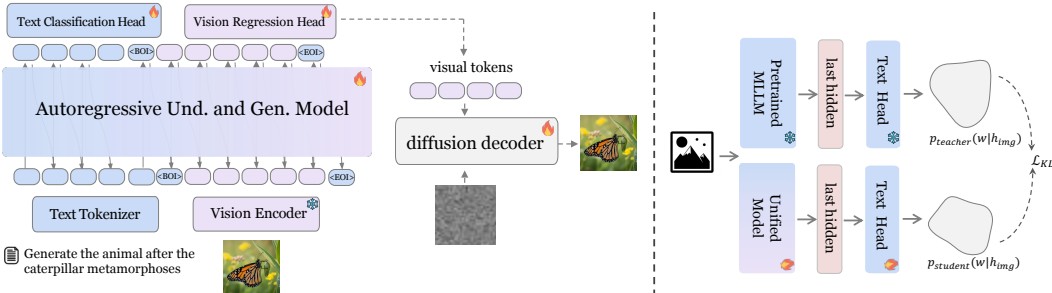

Figure 2: Method Overview. (Left) Our unified autoregressive model processes mixed text and visual tokens using two heads: a Text Classification Head for understanding and a Vision Regression Head to predict visual tokens for image synthesis. (Right) The Representation Consistency Loss prevents semantic drift by aligning our model's text output with a frozen pretrained model at visual token positions, ensuring semantic integrity during generation.

**Text Understanding Loss ($\mathcal{L}_{\mathbf{CE}}$):** When the next token to be predicted is text, the model computes a probability distribution over the vocabulary via a classification head (i.e., the language model's output embedding matrix). This process is supervised by the standard cross-entropy loss, which aims to maximize the log-likelihood of the correct text token. This constitutes the model's semantic learning objective.

$$\mathcal{L}_{\mathrm{CE}} = -\frac{1}{N_{text}} \sum \log p(T_i|V, T_{<i}) \tag{1}$$

**Image Generation Loss ($\mathcal{L}_{\mathbf{MSE}}$):** To endow the model with generative capabilities, when the next token to be predicted is a visual token, we require the model to directly regress its embedding vector. This process is supervised by the mean squared error loss, which minimizes the Euclidean distance between the predicted vector $\hat{v}_{j+1}$ and the ground-truth vector $v_{j+1}$. This forms the model's structural learning objective.

$$\mathcal{L}_{\mathrm{MSE}} = \frac{1}{N_{\mathrm{vision}}} \sum \|f(h_j) - v_{j+1}\|_2^2 \tag{2}$$

Here, $f(\cdot)$ represents the vision head that maps the hidden representation $h_j$ to the predicted visual token.

Additionally, we pretrain a separate diffusion model to act as an image decoder. This decoder takes the autoregressively generated sequence of visual tokens $\{\hat{v}_1, ..., \hat{v}_m\}$ as a conditional embedding to render the final pixel-level image during inference.

### 3.2 REPRESENTATION ALIGNMENT

**Representation Conflict.** When training the model by naively combining $\mathcal{L}_{\mathrm{CE}}$ and $\mathcal{L}_{\mathrm{MSE}}$, we observe a severe degradation in its image understanding capabilities post-training. This degradation is also reflected in a significant shift in its internal visual representations. As shown in Fig. 3, the original base model's text classification head can map visual tokens to reasonable textual semantics. However, after naive generative fine-tuning, the model's classification predictions for the same visual tokens devolve into nonsensical outputs, indicating a profound semantic shift.

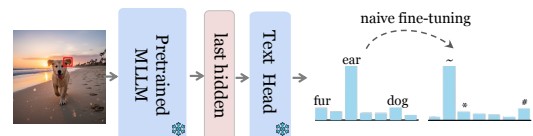

Figure 3: Semantic Drift. The rich semantic prediction for visual token from a pretrained MLLM collapses into a meaningless distribution after naive generative fine-tuning

To address this, we propose two key improvements, intervening at the decoder architecture and loss function levels.

**Vision head.** Prior works such as Emu (Sun et al., 2023) and Nexus-Gen (Zhang et al., 2025) use a linear vision head, creating a representational bottleneck. This directly propagates "semantically blind" pressure from the MSE gradient to the shared LLM representation $h_t$. We draw

inspiration from the text token prediction process, where the path from the hidden state to the final token (`hidden → logits (up-projection) → softmax(activation) → embedding lookup(down-projection)`) can be viewed as an implicit MLP regression. We therefore replace the this linear layer with a single-hidden-layer MLP head. This MLP serves as a nonlinear structural decoder, acting as a key-value memory(Geva et al., 2020) to map the high-level semantic $h_t$ to low-level visual token vectors. This design relaxes constraints on $h_t$; the MLLM no longer needs a linearly decodable semantic space, but only needs to output sufficient information to generate the next visual token. We find a well-pretrained MLP regression head significantly mitigates the decline in image understanding performance.

**Representational Consistency Loss ($\mathcal{L}_{\mathbf{KL}}$).** At positions where visual tokens are predicted, the model lacks explicit semantic supervision from the cross-entropy loss. We therefore introduce a representational consistency loss. The core idea is that when the model regresses a visual token, it must also simultaneously comprehend the textual semantics corresponding to that visual token. We achieve this through knowledge distillation, using a frozen, original base model as a teacher. During training, at each visual token position $j$, we enforce that the probability distribution $p_{\text{student}}(w|h_j)$ produced by our student model's text classification head should remain consistent with the distribution $p_{\text{teacher}}(w|h_j)$ produced by the teacher model. This consistency is measured by minimizing the KL-divergence between the two distributions.

$$\mathcal{L}_{\text{KL}} = \frac{1}{N_{\text{vision}}} \sum_{j \in \text{vision\_indices}} D_{\text{KL}}(p_{\text{teacher}}(w|h_j) \,||\, p_{\text{student}}(w|h_j)) \tag{3}$$

This loss term acts as a powerful semantic anchor, ensuring that as the model pursues structural reconstructability, its representations do not deviate from a meaningful semantic trajectory. In doing so, it aligns the two conflicting optimization objectives.

**Overall Training Objective.** The final loss function is a weighted sum of three loss terms:

$$\mathcal{L}_{\text{total}} = \mathcal{L}_{\text{CE}} + \lambda_{\text{MSE}}\mathcal{L}_{\text{MSE}} + \lambda_{\text{KL}}\mathcal{L}_{\text{KL}} \tag{4}$$

where $\lambda_{\text{MSE}}$ and $\lambda_{\text{KL}}$ are hyperparameters that balance the contributions of the different tasks.

## 3.3 MODEL TRAINING

### 3.3.1 DATA CURATION.

Our training corpus encompasses a diverse range of data types to ensure balanced development across all model capabilities. We carefully organize the public dataset to adapt to multi-task, which includes:

a) **Understanding Data:** To maintain and enhance comprehension abilities, we utilize the publicly available LLaVA-OV dataset(Li et al., 2024), which comprises 3.5M samples.

b) **Text-to-Image Data:** For training text-to-image generation, we filter the pre-training and fine-tuning datasets from BLIP-3o(Chen et al., 2025a), yielding 20M image-text pairs. Additionally, to ensure prompt diversity and high image quality, we leverage FLUX.1-dev(Labs, 2024) to regenerate 5M images from the LAION-COCO aesthetic dataset(Guangyil, 2022), then recaption by Qwen2.5 VL(Bai et al., 2025).

c) **Image Editing Data:** To cover various image editing tasks, we incorporate 1.2M samples from the UniWorld(Lin et al., 2025a).

### 3.3.2 PROGRESSIVE TRAINING RECIPE.

We repurpose the diffusion decoder from Nexus-Gen(Zhang et al., 2025), which is tasked with image reconstruction conditioned by visual token sequences $V$ encoded by the ViT. To smoothly inject generative capabilities into the pretrained Qwen2.5-VL(Bai et al., 2025) while maximally preserving its original understanding abilities, we adopt a curated multi-stage training recipe using a dynamic mixture of the curated data described above. Tab. 1 shows the configuration of different training stages in detail. We carefully adjusted the data and hyperparameters of each stage to achieve better performance.

Table 1: Detailed hyperparameters for different Training recipe

| Hyperparameters | Warm-up | Pretrain | SFT |
|---|---|---|---|
| Max learning rate | $1 \times 10^{-3}$ | $1.0 \times 10^{-4}$ | $1.0 \times 10^{-4}$ |
| Min learning rate | $1 \times 10^{-3}$ | 0 | $1.0 \times 10^{-5}$ |
| LR scheduler | Constant | Cosine decay | Cosine decay |
| Weight decay | 0.1 | 0.1 | 0.1 |
| Gradient norm clip | 1.0 | 1.0 | 1.0 |
| Optimizer | AdamW ($\beta_1 = 0.9, \beta_2 = 0.95, \epsilon = 1.0 \times 10^{-15}$) | | |
| Loss weight (CE : MSE : KL) | 0:1:0 | 1:1:0.01 | 1:1:0.01 |
| Warm-up ration | 5% | 5% | 5% |
| Training epoch | 2 | 2 | 1 |
| Batch size(all devices) | 2048 | 4096 | 4096 |
| Gen resolution (min and max long side) | (252,252) | (252, 252) | (252, 252) |
| Und resolution (min and max long side) | - | (56, 1024) | (56, 1024) |

**Stage 1: Vision Head Warm-up.** We freeze the full MLLM backbone and exclusively train the newly added MLP vision head. The training objective in this stage is solely the MSE loss ($\mathcal{L}_{\text{MSE}}$). This step is designed to sufficiently warm up the MLP, equipping it with a preliminary capability for structural prediction and thereby laying a solid foundation for subsequent joint training. This stage used 20M images with lower quality but diverse text-to-image data to sufficiently warm up the vision head, which we found to be crucial for subsequent performance.

**Stage 2: Full-Parameter Pretrain.** In this phase, we unfreeze all model parameters and conduct fine-tuning using a data mixture heavily weighted towards understanding tasks. The primary objective is to adapt the entire model to the new architectural components and tasks while strongly anchoring its original semantic understanding capabilities. We use high-quality 3.5M image understanding and 5M text-to-image data for this phase.

**Stage 3: Multi-task Supervised fine-tuning.** This is the final stage. We employ a comprehensive mixed dataset incorporating all data types (understanding, generation, and editing). The goal is to balance and jointly enhance all of the model's diverse task capabilities in a unified manner. This stage introduces 1.2M image editing data based on the Stage 2.

## 4 EXPERIMENTS

### 4.1 EXPERIMENT SETUP

**Implementation details.** Our model is built upon the 7B Qwen2.5-VL (Bai et al., 2025) backbone and the Nexus-Gen (Zhang et al., 2025) diffusion decoder. For generation, we represent images as 81 visual tokens, following Zhang et al. (2025), while preserving native multi-resolution inputs for understanding tasks. We train the model using the DeepSpeed ZeRO-3 framework (Aminabadi et al., 2022). For stability and efficiency, we skip training steps with anomalous gradients (Liu et al., 2024a) and employ sequence packing (Krell et al., 2021) to maximize GPU throughput.

**Baselines.** We compare our model against three categories of state-of-the-art (SOTA) baselines: **1) Understanding-Only MLLMs** like Qwen2.5-VL (Bai et al., 2025) and LLaVA-OV (Li et al., 2024), to assess the preservation of understanding abilities; **2) Generation-Only Models** such as SDXL (Podell et al., 2023) and Flux.1 dev (Labs, 2024), to benchmark image synthesis quality; and **3) Unified Models** spanning the cascaded, parallel, and monolithic paradigms (Sec. 2.1) for a direct, holistic comparison.

### 4.2 RESULTS

**Text-to-Image Generation** We conduct a comprehensive quantitative evaluation of our model's text-to-image (T2I) generation performance on the established GenEval benchmark(Ghosh et al., 2023). As presented in Tab. 2, the quantitative results indicate that our model achieves highly competitive performance on this benchmark. Firstly, it surpasses previous monolithic unified models and demonstrates competitive performance against the latest counterparts, establishing a new performance benchmark for this technical approach. Furthermore, our model exhibits capabilities comparable to recent unified models that employ more complex architectures and is superior to several strong

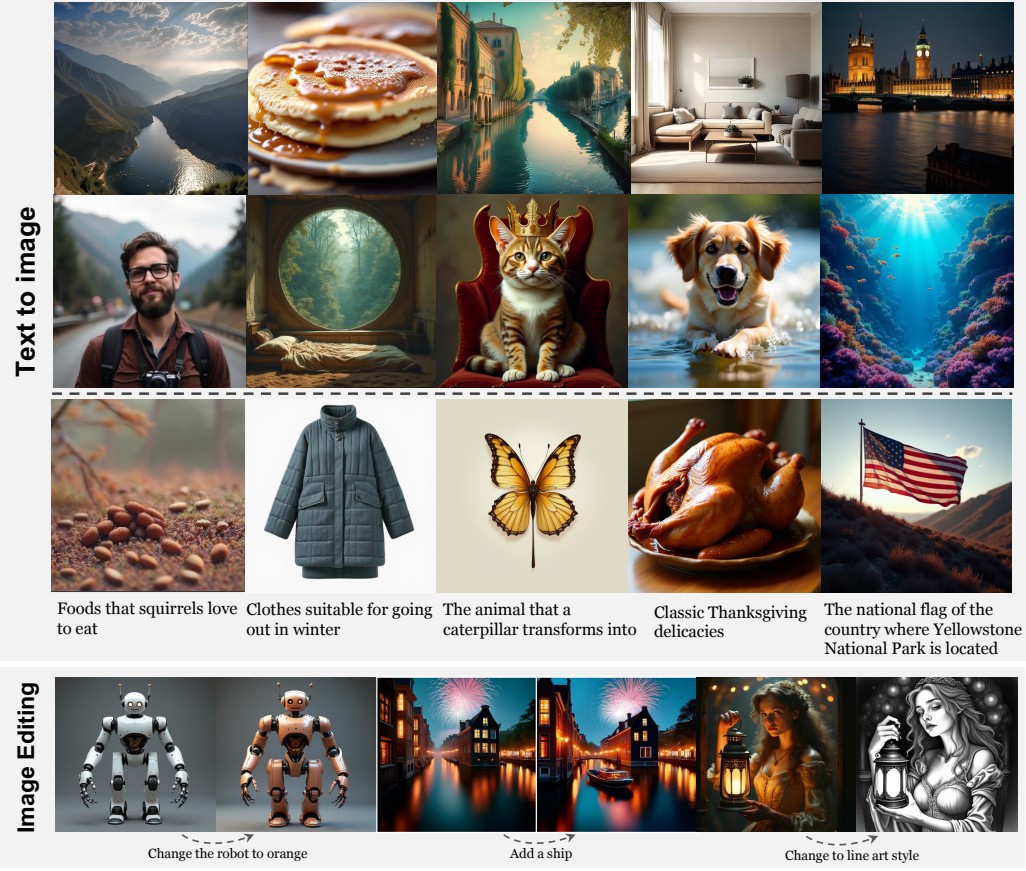

Figure 4: Overview of image generation capabilities. Our model can not only generate images from prompts (first two rows), but also generate images based on world knowledge reasoning (third row). In addition, it has certain local editing and global editing capabilities (bottom row).

generation-only baselines. This result provides strong evidence that the monolithic architecture, when optimized via our proposed framework, is fully capable of achieving top-tier generative quality without compromising its architectural simplicity and conversational fluidity.

Beyond the quantitative metrics, Fig. 4 provides a qualitative showcase of our model's synthesis capabilities. The examples in the first two rows demonstrate high prompt fidelity, with the model accurately rendering specified object compositions and complex scene layouts. Moreover, the model showcases impressive generalization when processing abstract prompts that require real-world knowledge, testifying to the effective synergy between the MLLM's reasoning core and the visual generation module(third row). More detailed results can be found in Appendix Fig. 6.

**Image Understanding** We evaluated our model's multimodal understanding performance across five comprehensive benchmarks: MMBench(Liu et al., 2024c), MMStar(Chen et al., 2024a), MMVet(Yu et al., 2023), SEED-Bench(Li et al., 2023), and RealWorldQA(X.AI, 2024). As presented in Tab. 3, the results show that our model not only largely preserves the strong understanding capabilities of its base model, Qwen2.5-VL, but also significantly outperforms all prior monolithic unified models. This key result validates that our proposed framework successfully enables image generation while effectively mitigating catastrophic forgetting, thereby overcoming the critical performance trade-offs that have previously hindered the development of monolithic approaches.

**Image Editing** Following multi-task SFT, our model also demonstrates certain image editing capabilities (Fig. 4, bottom row), performing both local(object add) and global(color change/style transfer) instruction-guided edits while preserving high fidelity in non-edited regions.

Table 2: Evaluation of text-to-image generation ability on GenEval benchmark, following Bagel(Deng et al., 2025) we use LLM rewriter.

| Method | Single | Two | Counting | Colors | Position | Color Attr. | Overall ↑ |
|---|---|---|---|---|---|---|---|
| *Gen. Only Models* | | | | | | | |
| SDXL(Podell et al., 2023) | 0.98 | 0.74 | 0.39 | 0.85 | 0.15 | 0.23 | 0.55 |
| DALLE3(Betker et al., 2023) | 0.96 | 0.87 | 0.47 | 0.83 | 0.43 | 0.45 | 0.67 |
| SD3-medium(Esser et al., 2024) | 0.99 | 0.94 | 0.72 | 0.89 | 0.33 | 0.60 | 0.74 |
| FLUX.1-dev(Labs, 2024) | 0.98 | 0.93 | 0.75 | 0.93 | 0.68 | 0.65 | 0.82 |
| *Unified Models* | | | | | | | |
| **Cascaded Architectures** | | | | | | | |
| OmniGen2(Wu et al., 2025b) | 0.99 | 0.96 | 0.74 | 0.98 | 0.72 | 0.75 | 0.86 |
| MetaQuery(Pan et al., 2025) | - | - | - | - | - | - | 0.80 |
| BLIP3-o(Chen et al., 2025a) | - | - | - | - | - | - | 0.84 |
| UniWorld-V1(Lin et al., 2025a) | 0.99 | 0.93 | 0.81 | 0.89 | 0.74 | 0.71 | 0.84 |
| **Parallel Architectures.** | | | | | | | |
| Mogao(Liao et al., 2025) | 1.00 | 0.97 | 0.83 | 0.93 | 0.84 | 0.80 | 0.89 |
| BAGEL(Deng et al., 2025) | 0.98 | 0.95 | 0.84 | 0.95 | 0.78 | 0.77 | 0.88 |
| **Monolithic Architectures** | | | | | | | |
| Janus-Pro(Chen et al., 2025b) | 0.99 | 0.89 | 0.59 | 0.90 | 0.79 | 0.66 | 0.80 |
| Emu3(Wang et al., 2024) | 0.99 | 0.81 | 0.42 | 0.80 | 0.49 | 0.45 | 0.66 |
| Show-o2(Xie et al., 2025) | 1.00 | 0.87 | 0.58 | 0.92 | 0.52 | 0.62 | 0.76 |
| Chameleon(Team, 2024) | - | - | - | - | - | - | 0.39 |
| Orthus(Kou et al., 2024) | - | - | - | - | - | - | 0.58 |
| **Ours** | 1.00 | 0.86 | 0.66 | 0.93 | 0.69 | 0.77 | 0.82 |

Table 3: Comparison with state-of-the-art methods on multimodal understanding benchmarks.

| Method | Base MLLM | MMB↑ | MMStar↑ | MMVet↑ | SEED↑ | RWQA↑ |
|---|---|---|---|---|---|---|
| *Und. Only Models* | | | | | | |
| Qwen2.5 VL(Bai et al., 2025) | - | 79.1 | 63.9 | 67.1 | 79.5 | 68.5 |
| LLaVA-OV(Li et al., 2024) | - | 80.8 | 61.7 | 57.5 | 75.4 | 66.3 |
| *Unified Models* | | | | | | |
| **Cascaded Architectures** | | | | | | |
| OmniGen2(Wu et al., 2025b) | Qwen2.5-VL 7B | 79.1 | 63.9 | 67.1 | 79.5 | 68.5 |
| MetaQuery(Pan et al., 2025) | Qwen2.5-VL 7B | 79.1 | 63.9 | 67.1 | 79.5 | 68.5 |
| BLIP3-o(Chen et al., 2025a) | Qwen2.5-VL 7B | 79.1 | 63.9 | 67.1 | 79.5 | 68.5 |
| UniWorld-V1(Lin et al., 2025a) | Qwen2.5-VL 7B | 79.1 | 63.9 | 67.1 | 79.5 | 68.5 |
| **Parallel Architectures.** | | | | | | |
| Mogao(Liao et al., 2025) | - | 75.0 | - | - | 74.6 | - |
| BAGEL(Deng et al., 2025) | Qwen2.5-VL 7B | 85.0 | - | 67.2 | - | - |
| **Monolithic Architectures** | | | | | | |
| Janus-Pro(Chen et al., 2025b) | DeepSeek-LLM 7B | 79.2 | - | 50.0 | 72,1 | - |
| Emu3(Wang et al., 2024) | EmuChat 7B | 58.5 | - | 37.2 | 68.2 | 57.4 |
| Show-o2(Xie et al., 2025) | Qwen2.5-7B | 79.3 | 56.5 | - | 69.8 | - |
| Chameleon(Team, 2024) | From scratch 7B | 35.7 | - | 8.3 | - | - |
| Orthus(Kou et al., 2024) | Chameleon-7B | - | - | - | - | - |
| **Ours** | Qwen2.5-VL 7B | 83.7 | 64.2 | 64.5 | 78.1 | 67.4 |

## 4.3 EMERGENT CAPABILITIES OF THE UNIFIED REPRESENTATION

Beyond quantitative benchmarks, our unified training framework gives rise to several zero-shot emergent capabilities. These abilities, not explicitly present in the training data, serve as evidence for the generalization power of the learned unified representation.

**Zero-shot Interleaved Generation and Understanding.** The model can engage in multi-round dialogue that seamlessly interleaves generation and understanding (Fig. 5, first row). It can generate an image and immediately answer questions about its own creation, despite being trained only on decoupled single-turn generation and multi-turn understanding data. We attribute this zero-shot capability to the alignment of text and image representations; the model treats its own generated visual tokens as understandable context for the next round.

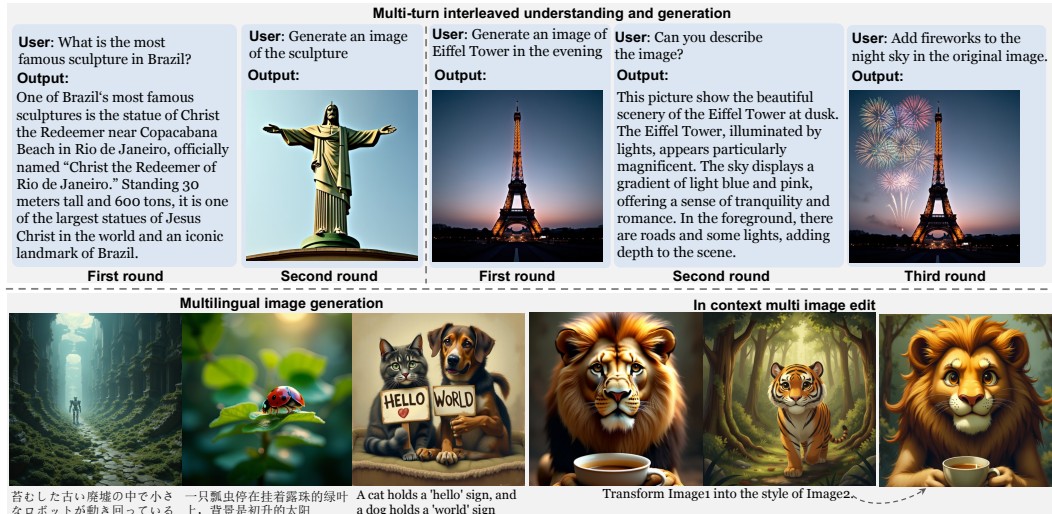

Figure 5: The zero-shot emergent capabilities.

**Cross-lingual Generalization.** Trained exclusively on English text-to-image data, the model exhibits zero-shot cross-lingual generation from prompts in other languages like Chinese and Japanese(Fig.5, second row, left). This capability is inherited from the powerful multilingual alignment already present in the Qwen2.5-VL backbone and successfully extended to the generative modality by our framework, showcasing robust generalization.

**Contextual Multi-image Editing.** Similarly, the model generalizes from its training on single-image editing data to performing contextual edits across multiple images in a single turn (Fig.5, second row, right). This underscores the strong compositional properties endowed by the unified representation space, allowing learned skills to be applied to more complex, novel scenarios.

## 4.4 ABLATION STUDIES

We systematically investigate the impact of two core components: the vision head architecture and our proposed representation consistent loss $\mathcal{L}_{KL}$. The results are presented in Tab.4.

**Impact of Vision Head Architecture and Warm-up.** The vision head is responsible for decoding the MLLM's hidden states into visual tokens. We compare our proposed MLP head against two alternatives: a simple Linear layer and a more complex Q-Former(parameter-matched with MLP). When warm-up vision head with a relatively small dataset of 5M samples, the Q-Former (Config C: 77.3 MMB) shows a slight advantage in understanding metrics over the MLP head (Config E: 76.4 MMB). However, as we scale the head Stage-1 data to

Table 4: **Ablation Study: Impact of Vision Head and Training Strategies.**

| Config | Vison head | $\mathcal{L}_{KL}$ | Stage 1 Data | Understand↑ | | Gen.↑ |
|---|---|---|---|---|---|---|
| | | | | MMB | MMS | GenEval |
| *Baselines and Alternatives* | | | | | | |
| A | Linear | ✗ | 5M | 71.6 | 54.3 | 0.62 |
| B | Linear | ✗ | 20M | 71.5 | 56.8 | 0.65 |
| C | Q-Former | ✗ | 5M | 77.3 | 60.7 | 0.75 |
| D | Q-Former | ✗ | 20M | 78.6 | 59.6 | 0.77 |
| *Our Proposed Approach* | | | | | | |
| E | MLP | ✗ | None | 74.1 | 57.5 | 0.73 |
| E | MLP | ✗ | 5M | 76.4 | 59.3 | 0.76 |
| F | MLP | ✗ | 20M | 79.8 | 61.0 | 0.79 |
| **G** | **MLP** | ✓ | **20M** | **83.7** | **63.2** | **0.82** |

20M samples, the MLP head's performance (Config F) not only closes the gap but clearly surpasses that of the Q-Former (Config D) across all metrics. This suggests that while Q-Former may be more sample-efficient initially, the MLP exhibits superior scaling properties and generalization potential. It underscores our conclusion that a sufficiently warm-up vision head is critical to unlocking the full capabilities of the monolithic framework.

**Efficacy of $\mathcal{L}_{KL}$.** The second key component of our framework is the $\mathcal{L}_{KL}$, designed to mitigate the semantic-structure conflict. By comparing our final model (Config G) with its direct counterpart trained without this loss (Config F), the inclusion of $\mathcal{L}_{KL}$ yields substantial and consistent gains across the board: a remarkable +3.9 point increase on MMBench and +2.2 on MMStar for understanding,

coupled with a solid +3.0 improvement in the GenEval score for generation. This concurrent enhancement of both understand and generative tasks provides strong empirical evidence for our central hypothesis: the $\mathcal{L}_{KL}$ effectively aligns the representations required for both tasks, ensuring that the model maintains semantic fidelity even as it learns to perform structural reconstruction.

## 5 CONCLUSION

We tackled the core challenge of unifying multimodal understanding and generation in monolithic autoregressive models, where a semanticstructural conflict often leads to catastrophic forgetting. To address this, we introduced a framework of Decoupling and Alignment, combining a non-linear vision head, a representation consistency loss, and a progressive training recipe. Our approach enables a purely monolithic model to preserve strong understanding while achieving competitive generation and editing performance, surpassing prior monolithic designs and rivaling more complex unified models. These results demonstrate that monolithic autoregression, when carefully aligned, is a simple yet effective path toward integrated multimodal intelligence.

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

## A APPENDIX

### A.1 LIMITATIONS AND FUTURE WORK

**Limitations** While our model achieves competitive performance in both understanding and generation within a purely monolithic and autoregressive framework without separate image encoders or task-specific parameters, this architecture exhibits notable limitations. The primary challenge lies in image editing tasks that require high fidelity to the source image. The visual representations fed to the diffusion decoder are derived from a semantic-rich vision encoder. Consequently, these representations, while excellent for generation, lack the fine-grained, low-level structural details necessary for precise, consistency-preserving edits. Although this can be partially mitigated through engineering solutions, such as concatenating low-level VAE tokens from the source image into the diffusion decoder, this approach introduces considerable complexity. For instance, in a multi-turn dialogue, determining which image a user intends to edit becomes a non-trivial challenge. The root of this issue lies in the absence of an image tokenizer that excels at both semantic understanding and pixel reconstruction. As the research community progresses towards this goal, our proposed framework is designed to seamlessly integrate such future advancements.

A second limitation pertains to generative diversity. Since visual tokens are predicted via regression, the generation process is deterministic and lacks the inherent stochasticity of sampling-based methods. We contend that this is not a major drawback. A potential solution is to structure the training data such that the model learns to take a concise user prompt and generate a more descriptive, elaborated prompt, which then conditions the image output. Diversity can then be achieved by sampling from the distribution of these elaborated prompts, effectively leveraging the language model's generative variety.

**Future Work** The primary advantage of the monolithic autoregressive architecture is its natural ability to support multi-turn, interleaved understanding and generation. Encouragingly, our model demonstrates this capability as an emergent property, even without being trained on relevant data. However, this conversational proficiency is still nascent and requires further development. Our immediate future work will focus on constructing a dedicated dataset for interleaved multimodal dialogue and subsequently fine-tuning the model to explicitly enhance these interactive capabilities.

### A.2 DATASET CONSTRUCTION DETAILS

In addition to leveraging publicly available datasets, we constructed a high-quality synthetic dataset of 5 million text-to-image pairs to enhance our training corpus. This dataset was generated through a carefully designed three-stage pipeline to ensure both conceptual diversity and high image-text alignment.

1. **Initial Prompt Generation:** To create a diverse collection of prompts covering a wide range of visual concepts, we first employed the Qwen2.5-VL 7B model(Bai et al., 2025) to recaption images from the LAION-COCO aesthetic dataset(Guangyil, 2022).

2. **Image Synthesis:** Next, these generated captions were used as prompts for the FLUX.1-dev(Labs, 2024) model to synthesize a new, high-quality set of images.

3. **Final Recaptioning:** Finally, to ensure strong text-image consistency, we performed a second recaptioning step. We used Qwen2.5-VL 7B once more to generate clean and accurate descriptions for the newly synthesized images.

This pipeline ensures that our final dataset possesses both rich visual diversity and robust image-text alignment. The training data for generative tasks follows a conversational format, similar to that used for understanding tasks. An example is provided below.

---

**Data Format Example**

```
{
  "id": 0,
  "image": "./000.jpg",
  "conversations": [
    {
      "from": "human",
      "value": "Can you make an image of an ancient library hidden
                in a deep forest, where bookshelves reach the canopy."
    },
    {
      "from": "gpt",
      "value": "<image>"
    }
  ]
},
```

---

To enhance the model's robustness to natural language, the human-provided prompts in the "conversations" array were constructed from a diverse set of templates. These templates cover a range of interaction styles, including polite requests (e.g., "I'd like to see a drawing of..."), direct commands (e.g., "Generate an image of..."), and questions (e.g., "Can you produce a picture of..."). This variety ensures the model is not overfitted to a single command structure.

### A.3 EXPLORATION OF AN ALTERNATIVE APPROACH

In an alternative approach, we explored leveraging features from multiple layers of the MLLM backbone, motivated by the hypothesis that different layers capture visual-semantic information at varying levels of granularity. We implemented a mechanism to compute a learnable, weighted average of the hidden states from all layers, like SENet(Hu et al., 2018). This aggregated feature vector was then passed to the vision regression head.

Interestingly, this method achieved a lower final convergence loss during training compared to our baseline approach, which uses only the final layer's representation. Despite this promising training dynamic, the resulting model exhibited a complete failure to generalize during inference, rendering it incapable of performing text-to-image synthesis. We hypothesize that this phenomenon is attributable to the compounding of prediction errors during autoregressive inferencea discrepancy between training on ground-truth embeddings and inferencing on self-generated ones. It is plausible that the minor errors in each predicted token are propagated and amplified through the network's successive layers. Our weighted-averaging mechanism may then inadvertently aggregate and exacerbate these errors from across the network's depth, leading to a catastrophic breakdown of the generative process.

### A.4 VISION HEAD ARCHITECTURE

The vision head is implemented as a single-hidden-layer Multilayer Perceptron (MLP). We utilize the GELU activation function and found empirically that setting the intermediate layer's dimension to eight times that of the input dimension yielded a strong balance of performance and efficiency. Further expansion of this intermediate dimension offered only marginal performance gains.

In our ablation studies (see Section 4.4), this MLP architecture was benchmarked against a Q-Former of a comparable parameter count. The Q-Former, which consisted of stacked self-attention layers, was configured to use a single query token to autoregressively predict the subsequent visual token.

### A.5 MORE QUALITATIVE RESULTS

Fig.6 presents a diverse array of images generated by our model. These results highlight the model's capability to produce high-fidelity images across a range of resolutions, all closely adhering to their corresponding textual descriptions.

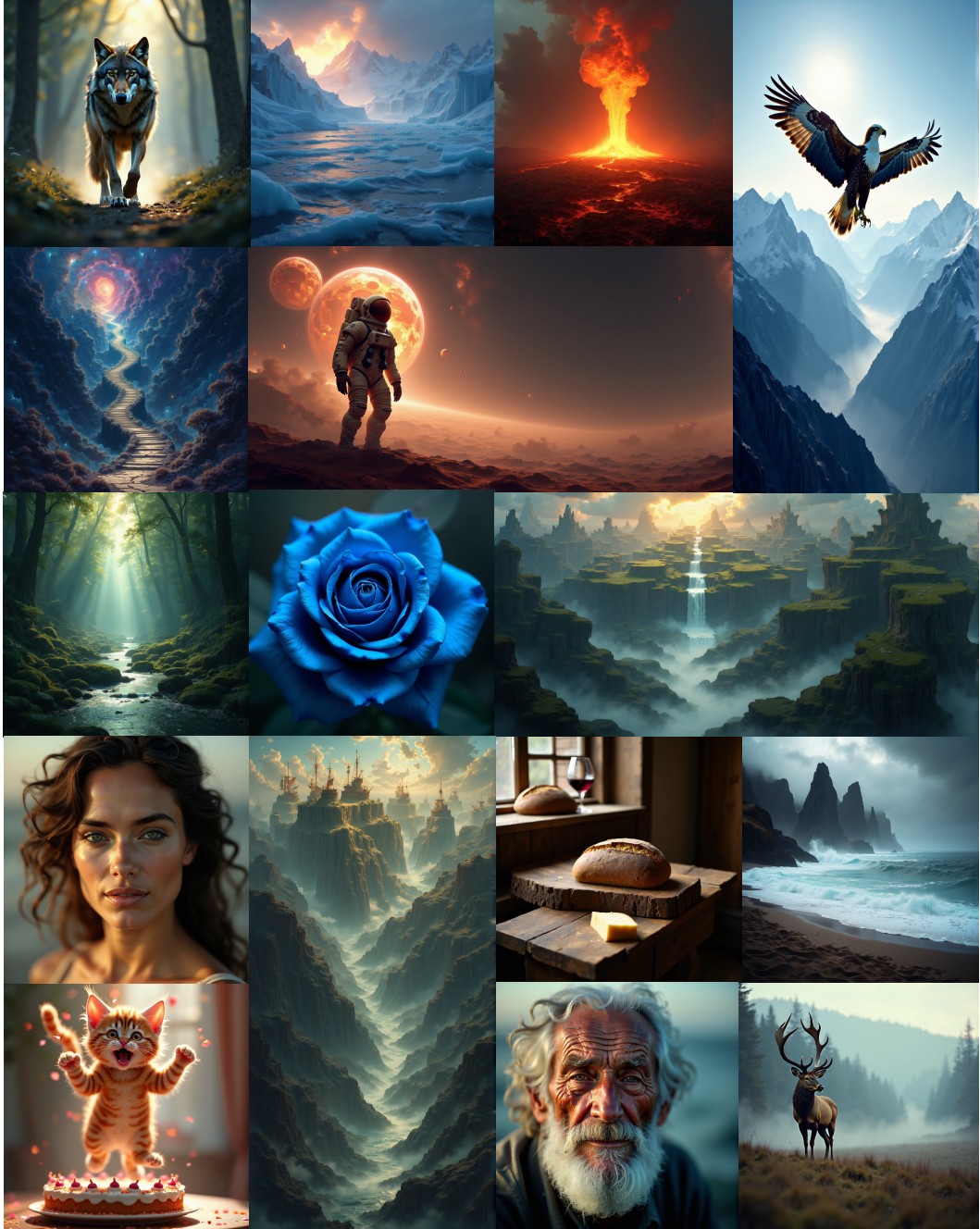

Figure 6: Visualization results of Ours model at various resolutions

### A.6 STATEMENTS

**Broader Impacts** While our work can enhance creative tools and human-computer interaction, we acknowledge the significant societal risks. The technology could be exploited to generate

misinformation, and the model may perpetuate societal biases learned from web-scale training data. We encourage the community to focus on developing robust safeguards and mitigation strategies for these potential harms.

**Ethical Statement**    We acknowledge the ethical risks of misuse and bias inherent in this work. To promote transparency and enable further research into model safety and fairness, we commit to the full open-sourcing of our code, models, and newly curated data. We advocate for the responsible development and deployment of this technology within the research community.

**Reproducibility Statement.**    To ensure full reproducibility, we commit to making our entire research pipeline publicly available. We will release all training code, curated dataset files, and final model checkpoints upon publication. Our model is built upon the publicly accessible Qwen2.5-VL backbone and the diffusion decoder from Nexus-Gen. All experiments were conducted using publicly available datasets, including LLaVA-OV, BLIP-30, LAION-COCO, and UniWorld, as detailed in the methodology section.

**LLM usage statement**    A large language model was utilized solely as a writing assistant to help polish the manuscript and improve the clarity and fluency of the English prose. The LLM was not involved in any core aspects of the scientific work, including the formulation of the initial hypothesis, experimental design, data analysis, or the derivation of our findings and conclusions. All intellectual contributions were made exclusively by the authors.

