# OpenReview forum: "ORION: Decoupling and Alignment for Unified Autoregressive Understanding and Generation"
_ICLR.cc/2026/Conference — ICLR 2026 Poster_

### Official Review · Reviewer_QFw6 · 2025-10-28

**Soundness:** 3
**Presentation:** 4
**Contribution:** 2
**Rating:** 4
**Confidence:** 5

**Summary:**

This paper introduces ORION, a unified framework that resolves the conflicts between understanding and generation through two decoupled vision heads for semantical and generative objectives separately.
The core contribution is:
1. The authors explicitly define the "semantic-structural representation conflict" in unified autoregressive MLLMs.
2. To resolve the conflict, the authors decouple semantic-preserving and low-level generation objectives by two vision heads:
+ The semantic-preserving is realized by aligning the semantic prediction distribution using the raw language heads with a frozen MLLM teacher of the same initial weights, with a proposed Representational Consistency Loss, i.e., $\mathcal{L}_{KL}$.
+ The low-level generation objective is realized by aligning the outputs of an additional vision head with the input embeddings of the MLLM, i.e., $\mathcal{L}_{MSE}$.
3. The authors dedicatedly curated a progressive training pipeline to warm up the vision heads and conduct full-parameter training.

The authors conduct experiments on several benchmarks and show that ORION can achieve SOTA generation ability while preserving its base MLLM's understanding ability.

**Strengths:**

1. The paper writing is very clear and easy to follow.
2. The figures are easy to read.
3. All designs, i.e., the vision regression head and the $\mathcal{L}_{KL}$, are rationalized well.
4. Promising results on both understanding and generation benchmarks.

**Weaknesses:**

1. **Incremental to SEED-X without discussion**.
This work adopts nearly the same architecture as SEED-X[1] without discussing the differences. Please correct me if I misunderstand something:
+ Both SEED-X and ORION receive native tokenized texts and ViT embeddings as the inputs.
+ For the text side, they are exactly the same, i.e., supervised by the Next-Token-Prediction objective.
+ For the vision side
    + SEED-X regresses the $N$ learnable queries with $N$ ViT embeddings using MSE to foster generation ability. They ignore the raw $N$ corresponding vision tokens because at that time, MLLMs are not well-developed, and the backbone is purely text-based, i.e., Llama-2. Therefore, the vision tokens do not have a decodable semantic space.
    + Since this work adopts a pre-trained MLLM as the backbone, the $N$ vision tokens become meaningful (decodable), and the learnable queries are no longer needed. Therefore, ORION switches to directly regress the $N$ vision tokens using the same MSE loss.
+ Both SEED-X and ORION fed vision outputs to a diffusion decoder to produce pixel output.

I recognize the contribution of the naturally extended $\mathcal{L}_{KL}$ objective because the vision tokens are decodable under the MLLM-as-backbone scenario. But for other parts, it looks like simply switching the regression target from $N$ queries (SEED-X) to $N$ raw vision outputs (ORION), which seems a bit incremental to SEED-X.

[1] SEED-X: Multimodal Models with Unified Multi-granularity Comprehension and Generation

2. **Overclaimed contribution.**
I do not agree with the authors' claim:

*"Conceptually, we are the first to identify and define the “semantic-structural representation conflict” in unified autoregressive MLLMs".*

There is a semantic-structural representation conflict between understanding and generation has become a consensus in the community. Several well-known works have already raised such an issue. For example, in Janus[1] (which is not cited by this paper):

```
The output of understanding task not only involves extracting information from images but also involves complex semantic reasoning. Therefore, the granularity of the vision encoder’s representation tends to mainly focus on high-dimensional semantic representation. By contrast, in visual generation tasks, the main focus is on generating local details and maintaining global consistency in the image. The representation in this context necessitates a low-dimensional encoding that is capable of fine-grained spatial structure and textural detail expression. Unifying the representations of these two tasks within the same space will lead to conflicts and trade-offs.
```

Many other papers regarding unified models have similar statements, so the authors should not claim they are **the first one** to discover it.

[1] Janus: Decoupling Visual Encoding for Unified Multimodal Understanding and Generation

3. **Inaccurate description of Cascaded Architectures.**

In L115-L116, the authors comment:

*The MLLM acts as a text encoder to guide image synthesis. While leveraging specialized models, this design cannot comprehend its own visual outputs, precluding multi-turn interaction.*

But many MLLM-based unified models receive multimodal input instead of pure text. Though some query-based works, e.g., MetaQuery, cannot directly understand the query, they indeed can understand the output images. By combining the output image with previous conversations and new user instructions as a multimodal input, they can also realize multi-turn interaction.

4. **Misclaimed architecture type.**

The authors claim ORION is a monolithic autoregressive model, but the generation requires a separately trained diffusion model, i.e., Nexus-Gen. From Figure 1, there is no difference between the Cascaded Architectures (Left) and the Monolithic Autoregressive Models (Right). From my perspective, true Monolithic Autoregressive Models should not rely on external diffusion models, e.g., Transfusion[1] and Show-o2[2]. ORION should be categorized as a Cascaded Architecture.

[1] Transfusion: Predict the Next Token and Diffuse Images with One Multi-Modal Model

[2] Show-o2: Improved Native Unified Multimodal Models

5. **Does not truly resolve the conflicts.**

+ From Table 3, we can observe that compared to Qwen2.5-VL-7B, the performance on most benchmarks, i.e., MMVet, SEED, and RWQA, shows a clear degradation, indicating that the semantic-structural representation conflict is not truly resolved.
+ From Figure 4 Image Editing part, the results are quite similar to those of MetaQuery, where the objects and backgrounds are significantly altered. For instance, the background of the robot becomes much brighter, and the robot's shape is changed. For the second image, the buildings on the left are totally different after editing. This evidence suggests that the learned representations do not truly preserve low-level details; instead, they still focus on semantic accuracy, and the model is doing regeneration instead of editing.

6. **Potential benchmark hacking because of the usage of BLIP-3o training data**.

BLIP-3o training data is a highly controversial one in the community, because it directly distills the GenEval benchmark. Training on such a corpus and evaluating on GenEval is unfair for other baselines, e.g., Janus-Pro and Show-o2. Also, as a unified model, evaluating only on GenEval is not sufficient. Please consider evaluating on other benchmarks, including those non-semantic-oriented ones, to show the true generation quality.

**Questions:**

1. Can you add more discussion about the difference between your work and SEED-X?
2. Can you show results on:
+ Semantics-oriented benchmarks other than GenEval, for instance, WISE and DPG-Bench.
+ Non-semantics-oriented benchmarks, such as MJHQ-30K and MS-COCO, these benchmarks should reveal if ORION learns aesthetic-aligned representations from a low-level view.

**I am willing to raise my score based on other reviewers' comments as well as the rebuttal quality.**

---

> ### Author Response · Authors · 2025-11-24
> **Response to Reviewer QFw6 -- Part1**
>
> We sincerely thank the reviewers for their thorough and highly insightful comments on this paper. Thanks so much for taking the time and effort to review our paper. The key issues you raised regarding the unified model architecture design (such as the comparison with SEED-X, the definition of architectural boundaries, etc.) are right on point and have greatly inspired us. However, we noticed that there might be some misunderstanding concerning the core contributions of this paper. Below, we provide point-by-point responses to the concerns you raised, hoping to further clarify the original intent and contributions of our design.
>
> **W1 & Q1: Incremental to SEED-X without discussion. This work adopts nearly the same architecture as SEED-X without discussing the differences.**
>
> While we acknowledge the high-level architectural similarities with SEED-X as the reviewer mentioned, we never claimed our contribution lay in the model structure itself. However, we respectfully disagree with the characterization of our work as an "incremental contribution." Our core contribution lies in solving a fundamental training challenge in unified multimodal modeling. We elaborate on this below:
>
> 1. **Paradigm Shift: Fixed Queries vs. Native Sequences**
>
> SEED-X relies on a fixed number of Learnable Query Tokens to regress image features. Constrained by the fixed query count, it struggles to flexibly handle complex, variable-length interleaved image-text sequences. In contrast, ORION achieves the direct regression of Native Visual Tokens from the sequence, naturally supporting interleaved generation of arbitrary lengths.
>
> We argue that SEED-X did not adopt this paradigm not because of the MLLM capabilities at the time (contemporaneous InternVL 1.5 and the earlier Emu2 already utilized the mainstream CLIP-style ViT $\rightarrow$ LLM architecture), but may because training native autoregression in a unified single model is inherently difficult.
>
> 2. **Fundamental Challenge: Semantic-Structural Representation Conflict**
>
> As shown in Table 4 of our paper, simply adding a linear Vision Head to an MLLM for joint training results in a drastic degradation of understanding performance and poor generation quality. Even Emu2, which also shares a similar architecture to ours, was forced to **split into two separate models (Emu2-Chat for understanding and Emu2-Gen for generation) after joint pretraining**. This highlights the critical issue we address: there is a fundamental conflict between semantic understanding and structural generation within a shared representation space.
>
> 3. **Our Systematic Solution**
>
> ORION proposes a systematic solution to reconcile this conflict, Not an incremental contribution:
>
> - **MLP Vision Head:** We introduce a parameter-heavy MLP projection layer acting as a "continuous vocabulary" for Visual Tokens. This effectively alleviates the representation bottleneck caused by the Linear Head used in Emu2/SEED-X.
> - **Representational Consistency Loss ($\mathcal{L}_{KL}$):** This is the key to resolving the conflict. Through $\mathcal{L}_{KL}$, we enforce the MLLM's **Last Hidden State** to retain semantic information (for text prediction) while simultaneously encoding structural information (for image reconstruction). Essentially, the model learns "how to construct the embedding (Structure)" without forgetting "what the embedding means (Semantics)." This is a non-trivial, distinct design choice rather than a natural extension.
> - **Curated Training Strategy and Data Ratios:** We conducted extensive experiments to determine the optimal strategy. Our results demonstrate that the warm-up of the non-linear Head and a task-progressive, multi-stage training pipeline are crucial for balancing understanding, generation, and editing tasks within a single model.
>
> In summary, our contribution demonstrates **how to achieve good multi-task performance within a single, unified model under shared representations.** We provide a scalable, general-purpose training paradigm. As the community develops better unified vision encoders and pixel decoders, our training strategy can be seamlessly integrated. We will further deepen this discussion based on Section 2.3 (Related Work) in the revised version.

---

> ### Author Response · Authors · 2025-11-24
> **Response to Reviewer QFw6 -- Part2**
>
> **W2: Overclaimed contribution of "Semantic-Structural Conflict"**
>
> We apologize for the ambiguity in our phrasing and the oversight in missing the citation for Janus. We fully agree that  "understanding and generation require different nature of representations" has become a community consensus.
>
> However, we wish to clarify that while the underlying principles share similarities, the **specific locus of the conflict and the challenge in resolving it** differ significantly:
>
> - **Previous Works (e.g., Janus):** Argument implies that a single vision encoder struggles to balance reconstruction and understanding. Therefore, they adopt **Input-side Decoupling** via a dual vision encoder structure (using a CLIP-style ViT for semantic understanding and a VQ-VAE for low-level generation information).
> - **ORION:** Focuses on a **Monolithic Autoregressive** architecture. Here, the conflict occurs at the **Output-side**—specifically, the **Shared Last Hidden State** must simultaneously satisfy the contradictory demands of semantic classification (text) and structural regression (image).
>
> Therefore, our contribution is not merely identifying the general concept of "representation difference," but **identifying and resolving this specific conflict at the Shared Last Hidden State level within a unified autoregressive model**, a conflict that typically leads to catastrophic forgetting. We will cite Janus [1] and revise our claim as follows:
>
> > "Conceptually, as far as we know we are the first to systematically identify and define the 'semantic-structural representation conflict' that arises when the **shared last hidden state** of a unified autoregressive MLLM is simultaneously used for both semantic text prediction and structural image reconstruction."
>
> **W3: Inaccurate description of Cascaded Architectures**
>
> We acknowledge that cascaded models can achieve multi-turn interaction by re-encoding generated images back into the input. However, we argue this is not a "Native" interactive capability:
>
> - **Intrinsic Consistency versus. External Loop:** In a cascaded mode, the model does not "understand" what it has generated until the image is rendered into pixels and re-encoded. In contrast, ORION’s internal representations allow it to "understand" its generation within the semantic space **before image re-encoding.**
> - **Generalization:** Thanks to joint autoregressive prediction in a unified representation space, ORION exhibits emergent, endogenous interleaved generation and understanding capabilities (as shown in Fig. 5) without being explicitly trained on specific "generation-understanding" multi-turn data.
> - **Efficiency:** Cascaded architectures incur additional overhead for the generated image encoding/decoding, whereas ORION maintains the coherence of autoregressive inference.
>
> We will update the text to state more precisely: "Cascaded models rely on external re-encoding of outputs to achieve multi-turn interaction, rather than possessing Intrinsic Autoregressive Consistency."
>
> **W4: Misclaimed architecture type**
>
> This is a highly insightful academic discussion regarding the definition of "Monolithic" architectures. The reviewer suggests that the presence of a Diffusion Decoder classifies ORION as a cascaded model. Based on the functional roles of the components, we respectfully offer a different perspective:
>
> 1. **Semantic Planner versus. Image Detokenizer:** In ORION, the autoregressive MLLM acts as the "Semantic Planner," responsible for all logical reasoning, semantic planning, and visual token prediction. The Diffusion Decoder merely acts as an "**Image Detokenizer**," similar to a VAE Decoder, and does not participate in logical generation.
> 2. **Contrast with Cascaded Models:** In typical cascaded architectures (e.g., Qwen-Image), the MLLM only provides text embeddings, and the generative reasoning relies heavily on the subsequent diffusion model. In ORION, the Diffusion Decoder does not participate in content logic generation but is responsible only for reconstructing high-frequency details.
> 3. **Replaceability:** As demonstrated by recent research like RAE [1], CLIP-style ViT Embeddings paired with non-diffusion decoders can also achieve high-quality reconstruction. This proves that the Diffusion Decoder is not a core logical component of our architecture and could theoretically be replaced by non-diffusion decoders.
>
> Therefore, given that all multimodal planning and semantic adherence occur within the single MLLM backbone, we classify ORION as a monolithic autoregressive architecture.

---

> ### Author Response · Authors · 2025-11-24
> **Response to Reviewer QFw6 -- Part3**
>
> **W5: Does not truly resolve the conflicts.**
>
> **1. Regarding Image Understanding Benchmarks:**
>
> As shown in Table 3, although there is a slight degradation in some understanding metrics compared to the base model Qwen2.5-VL (e.g., MMVet 67.1 $\rightarrow$ 64.5), this is a common trade-off when integrating large-scale generation capabilities into a single model. The crucial point is that ORION significantly outperforms other unified architectures (such as Chameleon and Emu3), demonstrating that our approach balances this conflict better than existing solutions. The ablation studies in Table 4 further validate the effectiveness of our individual components.
>
> **2. Regarding Image Editing:**
>
> It is a community consensus that without VAE tokens containing low-level information at the input side (as seen in dual-encoder approaches like Janus/Bagel), maintaining high fidelity in editing is challenging. However, Dual Encoders introduce significant training complexity and instability. ORION utilizes a unified representation space with a Single ViT Encoder. While this presents challenges for pixel-level fidelity, it facilitates generalization (as shown in Fig. 5, ORION exhibits emergent interleaved generation and understanding capabilities without being explicitly trained on such multi-turn datasets).
>
> Furthermore, recent research RAE [1] indicates that CLIP-style representations are sufficient for high-quality reconstruction if the decoder is properly trained. To this end, we train a Diffusion Decoder specialized for editing during rebuttal, which takes the original image's QwenViT Embedding and the editing embedding predicted by ORION as inputs.
>
> Following the Nexus-Gen [4] protocol, we evaluated 1,000 held-out editing samples. Metrics include CLIP-T (text-image consistency), L1 Distance (pixel difference), and CLIP-O/DINO-O (image embedding similarity). As shown in **Table R1**, ORION achieves **competitive fidelity and instruction adherence compared to diffusion-based baselines**, validating our approach.
>
> | **Method**      | **CLIP-T ↑** | **L1 ↓**  | **CLIP-O ↑** | **DINO-O ↑** |
> | --------------- | ------------ | --------- | ------------ | ------------ |
> | InstructPix2Pix | 0.281        | 0.169     | 0.815        | 0.723        |
> | UltraEdit       | 0.310        | 0.149     | 0.841        | 0.729        |
> | OmniGen         | 0.317        | 0.143     | 0.865        | 0.753        |
> | **Ours**        | **0.319**    | **0.137** | **0.892**    | **0.828**    |
>
> **Table R1: Quantitative Image Editing Results**
>
> This demonstrates that the unified representations learned by ORION are robust and can be combined with more advanced unified Vision Encoders in the future.
>
> **W6 & Q2: Potential benchmark hacking because of the usage of BLIP-3o**
>
> Thank you for your rigorous review. We take the potential issue of data overlap with BLIP-3o very seriously. To ensure the fairness of our evaluation, we conducted the following supplementary experiments:
>
> 1. **Exclusion Experiment:** We excluded the BLIP-3o 60k data and retrained ORION. Following the evaluation protocol of BAGEL (using an LLM rewriter), the GenEval score remained at **0.81**. This strongly demonstrates that our high performance stems from model capability rather than data hacking.
> 2. **Additional Benchmarks:** As requested, we evaluated our model on non-semantic benchmarks: WISE (World Knowledge) and MJHQ-30K (Image Quality).
>
> | **Method**   | **Janus-Pro** | **MetaQuery-XL** | **Ours** |
> | ------------ | ------------- | ---------------- | -------- |
> | WISE [2]     | 0.35          | 0.55             | **0.52** |
> | MJHQ-FID [3] | 13.48         | 6.02             | **5.58** |
>
> **Table R4: Quantitative Results on WISE and MJHQ-FID**
>
> The results show that ORION is comparable to MetaQuery-XL on tasks requiring world knowledge reasoning (WISE) and performs better in terms of image aesthetic quality (MJHQ-FID).
>
>
>
> **We once again express our gratitude for the reviewer's detailed and insightful comments. We hope these clarifications address your concerns and would greatly appreciate it if you could consider raising the score. We will add the experiments and analysis in the rebuttal to our paper.**

---

> > ### Comment · Reviewer_QFw6 · 2025-11-24
> > **Response to the rebuttal**
> >
> > Thanks for providing the detailed rebuttal. Most of my concerns are resolved. Though I still hold a different view regarding the categorization of ORION (monolithic vs cascaded), I believe this differs from reviewer to reviewer, and I do not intend to be too harsh. Generally, I appreciate this paper, and I am willing to raise my score to 6 to show my recognition. Anyway, please carefully check that all statements are fair to avoid potentially inaccurate overclaim :)

---

> > > ### Author Response · Authors · 2025-11-24
> > >
> > > Dear Reviewer QFw6,
> > >
> > > Thank you very much for your timely feedback and for acknowledging our rebuttal. We truly appreciate your willingness to raise the score.
> > >
> > > Regarding the categorization of ORION (monolithic vs. cascaded), we respect your perspective and understand the nuances you pointed out. We will also strictly follow your advice to carefully proofread our statements in the final version to ensure fairness and avoid any potential overclaims.
> > >
> > > Thanks again for your constructive comments which have helped improve our paper.

---

### Official Review · Reviewer_YV3K · 2025-10-30

**Soundness:** 3
**Presentation:** 3
**Contribution:** 3
**Rating:** 6
**Confidence:** 4

**Summary:**

The paper proposes ORION, a unified multimodal framework that keeps a monolithic autoregressive backbone while attempting to reconcile a central tension between semantic fidelity for understanding and structural reconstructability for generation. The method introduces two key pieces. First, a non-linear MLP vision head decouples low-level regression pressure from the shared backbone. Second, a representation consistency loss uses a frozen teacher to align the student’s text-prediction distribution at visual token positions, aiming to prevent semantic drift during generative training. The approach is trained with a progressive three-stage recipe that warms up the vision head, then performs full-parameter pretraining with an understanding-heavy mix, and finally multi-task supervised fine-tuning that includes editing. Experiments report competitive text-to-image scores on GenEval and strong preservation of understanding across MMBench, MMStar, MMVet, SEED-Bench, and RealWorldQA, alongside ablations that attribute gains to the MLP head and the consistency loss.

**Strengths:**

The paper isolates a concrete failure mode in unified autoregressive models, namely semantic drift during generative fine-tuning, and frames it as a representation conflict. The proposed remedy is conceptually simple yet targeted. The MLP head reduces direct gradient pressure on shared representations, and the representation consistency loss anchors semantics where cross-entropy supervision is absent. The training curriculum is pragmatic and clearly described, including data composition and hyperparameters, which improves reproducibility. Reported results indicate that ORION narrows the gap between monolithic models and more complex unified systems, while maintaining multi-turn interleaving abilities. The ablations are helpful and show that both the warm-up and the consistency loss matter, with measurable gains on understanding and generation metrics.

**Weaknesses:**

First, the representation consistency loss depends on a frozen teacher distribution at visual token positions. The paper should clarify whether the teacher sees exactly the same tokenization and masking, and whether temperature scaling or label smoothing is used. Without that detail, it is hard to judge stability and potential confirmation bias, since the student is nudged toward the teacher’s pretraining priors.
Second, the training recipe relies on large data mixtures that include regenerated images and recaptions. The paper should detail data licensing, deduplication against evaluation sets, and the exact share of synthetic captions. This is important for fairness and for assessing potential overfitting to benchmarks.
Third, although the ablation table is useful, it collapses several factors at once. For example, Q-Former vs MLP comparisons change the warm-up corpus size, and the effect of the loss weights is not isolated. A factorial ablation on λ values and warm-up sizes would make the causal story more convincing.
Finally, some claims about emergent multi-turn and cross-lingual behavior are illustrated qualitatively. These are compelling, but a small, controlled quantitative probe would help, for instance by measuring understanding accuracy on the model’s own generations and by using non-English GenEval-style subsets.

**Questions:**

1. Teacher details for the consistency loss. What teacher is used exactly, and does it share weights with the initialization of the student backbone. Do you match logits with a temperature or probabilities directly. How sensitive are results to the loss weight λKL and to masking strategies at visual token positions.
2. Data governance. Please specify data sources, licenses, and any filtering for copyrighted or sensitive content. Clarify whether FLUX regenerated images and Qwen recaptions appear in the test distributions and how you prevented leakage into GenEval or understanding benchmarks.
3. Robustness and safety. Did you assess failure modes like hallucinated text in images, biased generations, or unsafe outputs during editing. Any guardrails applied during training or inference would be useful to report.

---

> ### Author Response · Authors · 2025-11-24
> **Response to Reviewer YV3K**
>
> We extend our gratitude for your insightful feedback and suggestions.
>
> **W1 & Q1: Details on Representation Consistency Loss ($\mathcal{L}_{KL}$)**
>
> - **Teacher Settings:** The teacher model is a frozen original Qwen2.5-VL. It shares the exact same tokenizer and input processing pipeline as the student model, ensuring that both models observe identical token sequences and masking patterns.
> - **Loss Formulation:** We employ a standard knowledge distillation paradigm. Specifically, we compute the KL Divergence between the student's logits and the teacher's logits at visual token positions. Both sets of logits are processed via Softmax with a temperature of $T=1.0$.
> - **Sensitivity Analysis:** We conducted a sensitivity analysis on the loss weight $\lambda_{KL}$ (see **Table R3**). The results reveal a trade-off: a low weight (0.005) provides insufficient semantic constraint, limiting understanding gains; a high weight (0.02) strengthens understanding but over-regularizes the model, degrading generation flexibility. We empirically found $\lambda_{KL}=0.01$ to offer the optimal balance.
>
> | **$\lambda_{KL}$** | 0    | 0.005 | 0.01 (Ours) | 0.02 |
> | ------------------ | ---- | ----- | ----------- | ---- |
> | **MMBench**        | 79.8 | 81.0  | **83.7**    | 84.1 |
> | **GenEval**        | 0.79 | 0.81  | **0.82**    | 0.78 |
>
> **Table R3: Sensitivity of $\mathcal{L}_{KL}$ Weight**
>
> **W2 & Q2: Data Governance and Contamination**
>
> - **Licensing Compliance:** All datasets used (LLaVA-OV, BLIP-3o, UniWorld) are open-source and have licenses that permit research use. The LAION-COCO images were regenerated using FLUX.1-dev, a model that allows its outputs to be used for research.
> - **Decontamination:** We have conducted strict checks on the training data, and there is no overlap with the benchmark test set. The recaptioning process only uses Qwen2.5-VL to process the training images, and there is no overlap with the benchmark test set.
>
> **W3: Clarification on Ablation Study**
>
> First, we clarify that the ablation studies in Table 4 of the main paper are **controlled experiments**. Comparisons such as *Config C vs. E* and *D vs. F* were conducted under identical Stage 1 data sizes and configurations, isolating the specific impact of the components without coupling factors.
>
> To further address your suggestion for factorial isolation, we performed an additional ablation designed to completely separate the contributions of the Head Type (Linear vs. MLP) and the Alignment Loss ($\mathcal{L}_{KL}$).
>
> As shown in Table R2:
>
> 1. **Isolation of Alignment:** Applying $\mathcal{L}_{KL}$ to a standard Linear head improves understanding (71.5 $\to$ 77.8), confirming that the alignment loss provides intrinsic benefits independent of the head architecture.
> 2. **Necessity of Decoupling:** The combination of **MLP + $\mathcal{L}_{KL}$** yields the highest performance, demonstrating that non-linear decoupling is necessary to break the bottleneck and fully unleash the potential of alignment.
>
> | **Configuration**  | **Vision Head** | **$\mathcal{L}_{KL}$** | **MMBench (Und.)↑** | **GenEval (Gen.)↑** |
> | ------------------ | --------------- | ---------------------- | ------------------- | ------------------- |
> | Linear Baseline    | Linear          | No                     | 71.5                | 0.65                |
> | Linear + Alignment | Linear          | Yes                    | 77.8                | 0.73                |
> | ORION (Ours)       | MLP             | Yes                    | **83.7**            | **0.82**            |
>
> **W4: Quantitative Probe for Emergent Capabilities**
>
> To quantify the claim of "emergent multi-lingual generation," we designed a controlled probe by **translating GenEval prompts into Chinese.**
>
> The model achieved a GenEval score of **0.75** with Chinese prompts. While slightly lower than the English score (**0.82**), this is a significant result given that the model was never trained on Chinese generation data. This quantitatively proves that the monolithic architecture successfully transfers semantic knowledge from the understanding module to the generation task in a zero-shot manner.
>
> **We hope these clarifications address your concerns and would greatly appreciate it if you could consider raising the score. We will add the experiments and analysis in the rebuttal to our paper.**

---

### Official Review · Reviewer_FMab · 2025-11-01

**Soundness:** 3
**Presentation:** 3
**Contribution:** 3
**Rating:** 6
**Confidence:** 4

**Summary:**

This paper proposes ORION, a unified autoregressive model that aims to reconcile the inherent conflict between semantic understanding and structural reconstruction in multimodal large language models (MLLMs). The core claim is that a "semantic-structural representation conflict" hinders monolithic architectures, which the authors address via "decoupling and alignment." Specifically, they introduce a non-linear MLP vision head to decouple structural prediction and a Representation Consistency Loss to align semantics during generation. The model, built on Qwen2.5-VL and a Nexus-Gen decoder, is evaluated on standard benchmarks, showing competitive results in both understanding and generation without task-specific parameters.

**Strengths:**

- Clearly formalizing the semantic-structural conflict in unified autoregressive models.
- The model is evaluated on a wide range of standard benchmarks for both understanding and generation.
- The paper is well-written and easy to follow.

**Weaknesses:**

- The paper fails to discuss and contrast its approach with other recent works that pursue a similar monolithic, autoregressive vision (such as GLaMM or Emu3).
- Presenting image editing as a result without quantitative evaluation.
- The ablation study does not full isolate the mechanisms of "decoupling" and "alignment."
- The paper lacks of discussion on training cost. The progressive three-stage training, while effective, appears computationally intensive.

**Questions:**

- Can you provide quantitative results on a standard image editing benchmark to support the claim "certain image editing capabilities."
- Can you design an ablation that separates  "decoupling" and "alignment?" For instance, applying the KL loss to a model with a linear head to see if "alignment" alone can compensate for poor "decoupling."
- Can you provide a discussion of the total training cost compared to simpler fine-tuning or other unified models (e.g., parallel architectures).

---

> ### Author Response · Authors · 2025-11-24
> **Response to Reviewer FMab**
>
> We sincerely appreciate the valuable feedback and constructive suggestions. Thanks so much for taking time and effort to review our paper.
>
> **W1: Missing discussion on similar monolithic models (GLaMM, Emu3)**
>
> Thank you for your insightful question. Discussing recent monolithic models strengthens our positioning. We will add the following discussion to the revision:
>
> - **Emu3:** Emu3 relies on **discrete Visual Quantization** for nexrt-token prediction. While elegant, we argue that discrete tokens often prioritize texture over semantics, making it difficult to co-evolve understanding and generation capabilities. Consequently, **Emu3's open-source release split into separate models**: **Emu3-Chat** for understanding and **Emu-Gen** for generation. In contfrast, ORION employs **continuous token regression** via a CLIP-style QwenViT. As shown in Table 3, this approach preserves understanding capabilities better than discrete methods, maintaining a truly unified monolithic architecture.
> - **GLaMM:** While GLaMM is a powerful unified model, it utilizes a Region-of-Interest mechanism focused primarily on grounding and segmentation. ORION differs by aiming for generalist "omni" capabilities—encompassing high-fidelity generation, editing, and understanding and via a single backbone without task-specific encoders.
>
> **W2 & Q1: Lack of quantitative evaluation on image editing**
>
> We apologize for the omission. As detailed in **General Response 1**, we trained a specialized Diffusion Decoder during rebuttal period. It takes two inputs: the source image embedding (via QwenViT, to ensure editing fidelity with the source image) and the target image embedding predicted by ORION, to decode the final editing image pixel.Following the Nexus-Gen protocol, we evaluated 1,000 held-out editing samples. Metrics include CLIP-T (text-image consistency), L1 Distance (pixel difference), and CLIP-O/DINO-O (image embedding similarity).
>
> As shown in **Table R1**, ORION achieves **competitive fidelity and instruction adherence compared to diffusion-based baselines**, validating our approach.
>
> | **Method**      | **CLIP-T ↑** | **L1 ↓**  | **CLIP-O ↑** | **DINO-O ↑** |
> | --------------- | ------------ | --------- | ------------ | ------------ |
> | InstructPix2Pix | 0.281        | 0.169     | 0.815        | 0.723        |
> | UltraEdit       | 0.310        | 0.149     | 0.841        | 0.729        |
> | OmniGen         | 0.317        | 0.143     | 0.865        | 0.753        |
> | **Ours**        | **0.319**    | **0.137** | **0.892**    | **0.828**    |
>
> **Table R1: Quantitative Image Editing Results**
>
> **W3 & Q2: Failure to isolate mechanisms of "decoupling" and "alignment"**
>
> Thank you for suggesting we isolate these factors. We conducted an ablation study fixing the Stage 1 data and other experimental settings, comparing a Linear Head vs. MLP (with and without $\mathcal{L}_{KL}$). The results in **Table R2** reveal two conclusions:
>
> - **Effectiveness of Alignment:** Adding $\mathcal{L}_{KL}$ to a standard Linear head also improves understanding (71.5 $\to$ 77.8), confirming the alignment  is intrinsically beneficial regardless of head architecture.
> - **Necessity of Non-linearity:** The combination of **MLP + $\mathcal{L}_{KL}$**  yields the highest performance, demonstrating that non-linear decoupling is essential to fully unleash the potential of alignment.
>
> | **Configuration**  | **Vision Head** | **$ \mathcal{L}_{KL} $** | **MMBench (Und.)↑** | **GenEval (Gen.)↑** |
> | ------------------ | --------------- | ---------------------- | ------------------- | ------------------- |
> | Linear Baseline    | Linear          | No                     | 71.5                | 0.65                |
> | Linear + Alignment | Linear          | Yes                    | 77.8                | 0.73                |
> | ORION (Ours)       | MLP             | Yes                    | **83.7**            | **0.82**            |
>
> **Table R2: Factor Ablation of $\mathcal{L}_{KL}$**
>
> **W4 & Q3: Lack of discussion on training cost compared to other models**
>
> We apologize for missing this discussion. ORION demonstrates superior efficiency in two aspects:
>
> 1. **Architectural Efficiency:** Unlike parallel architectures (e.g., BAGEL) that maintain separate parameters for understanding and generation, ORION shares the vast majority of parameters (the MLLM backbone), reducing deployment costs.
> 2. **Data Efficiency:** Our progressive training is highly cost-effective. Stage 1 (warm-up) uses the largest dataset (20M) but only trains the lightweight vision head. The full-parameter training in Stages 2 & 3 uses only 10M data. In stark contrast, BAGEL requires **2.6B (2665M)** data to achieve stable performance. ORION achieves unified capabilities with significantly lower data costs.
>
> **We hope these clarifications address your concerns and would greatly appreciate it if you could consider raising the score. We will add the experiments and analysis in the rebuttal to our paper.**

---

### Author Response · Authors · 2025-11-24
**General response to reviewers**

We sincerely thank all reviewers (**R1=FMab, R2=YV3K, R3=QFw6**) for their insightful and constructive feedback. We are encouraged that the reviewers found our formalization of the "semantic-structural representation conflict" clear **(R1, R2)**, our proposed remedy of decoupling and alignment "conceptually simple yet targeted" **(R2)**, and our training recipe "pragmatic" **(R1)**. We appreciate the recognition of our solid performance on benchmarks **(R1, R3)**. Below we respond to general questions raised by reviewers. We use **W** to abbreviate Weaknesses, **Q** to represent Questions.

**General Response 1: Quantitative Experiments on Image Editing (R1-W2&Q1, R3-W5)**

1. **Trade-off between Fidelity and Generalization**

As the community consensus that lacking VAE tokens (rich in low-level information) poses challenges to editing fidelity with the source image. Existing methods like Bagel [1] and Janus [2] use dual vision encoders (VAE + CLIP-style ViT). However, dual encoders force the model to adapt to two distinct input spaces, significantly increasing training instability and difficulty.

In contrast, ORION uses a unified single-ViT representation for generation, understanding, and editing. While this entails a trade-off in reconstruction fidelity, it enables superior **generalization**. As shown in Fig. 5, ORION exhibits emergent capabilities in **zero-shot interleaved generation and understanding without specific multi-turn training** thanks to the unified representation space.

2. **Feasibility of Single-Encoder Editing**

Recent work like RAE [3] validates that **CLIP-style embeddings can yield high-quality reconstruction** with a proper decoder. Inspired by this, we trained a specialized Diffusion Decoder during rebuttal period. It takes two inputs: the source image embedding (via QwenViT, to ensure editing fidelity with the source image) and the target image embedding predicted by ORION, to decode the final editing image pixel.

3. **Quantitative Results**

 We evaluated 1,000 held-out editing samples following NexusGen. Metrics include CLIP-T (text-image consistency), L1 Distance (pixel difference), and CLIP-O/DINO-O (image embedding similarity). As shown in **Table R1**, ORION achieves **competitive fidelity and instruction adherence compared to diffusion-based baselines**, validating our approach.

| **Method**          | **CLIP-T ↑** | **L1 ↓**  | **CLIP-O ↑** | **DINO-O ↑** |
| ------------------- | ------------ | --------- | ------------ | ------------ |
| InstructPix2Pix [5] | 0.281        | 0.169     | 0.815        | 0.723        |
| UltraEdit [6]       | 0.310        | 0.149     | 0.841        | 0.729        |
| OmniGen [7]         | 0.317        | 0.143     | 0.865        | 0.753        |
| **Ours**            | **0.319**    | **0.137** | **0.892**    | **0.828**    |

**Table R1: Quantitative Image Editing Results.**

How to train a unified visual encoder that can achieve both good reconstruction and understanding is an important issue in the field. Once the community has an excellent encoder, it can be combined with ORION to achieve better editing effects.

**General Response 2: Ablation Study on $ \mathcal{L}_{KL} $ (R1-W3&Q2, R2-W3)**

To isolate the impact of the alignment loss, we conducted an ablation study fixing the Stage 1 data and other experimental settings, comparing a Linear Head vs. MLP with and without $ \mathcal{L}_{KL} $. The results in **Table R2** reveal two conclusions:

- **Effectiveness of Alignment:** Adding $ \mathcal{L}_{KL} $ to a standard Linear head also improves understanding (71.5 $\to$ 77.8), confirming the loss alignment is intrinsically beneficial regardless of head architecture.
- **Necessity of Non-linearity:** The combination of **MLP + $ \mathcal{L}_{KL} $** yields the highest performance, demonstrating that non-linear decoupling is essential to fully unleash the potential of alignment.

| **Configuration**  | **Vision Head** | **$ \mathcal{L}_{KL} $** | **MMBench (Und.)↑** | **GenEval (Gen.)↑** |
| ------------------ | --------------- | ---------------------- | ------------------- | ------------------- |
| Linear Baseline    | Linear          | No                     | 71.5                | 0.65                |
| Linear + Alignment | Linear          | Yes                    | 77.8                | 0.73                |
| ORION (Ours)       | MLP             | Yes                    | **83.7**            | **0.82**            |
**Table R2: Factor Ablation of $ \mathcal{L}_{KL} $**

[1] Emerging Properties in Unified Multimodal Pretraining

[2] Janus: Decoupling Visual Encoding for Unified Multimodal Understanding and Generation

[3] Diffusion Transformers with Representation Autoencoders

[4] Nexus-Gen: A Unified Model for Image Understanding, Generation, and Editing

[5] InstructPix2Pix: Learning to Follow Image Editing Instructions

[6] UltraEdit: Instruction-based Fine-Grained Image Editing at Scale

[7] OmniGen: Unified Image Generation

---

### Author Response · Authors · 2025-12-02
**Kind Summarization of Author-Reviewer Discussion**

Dear Area Chair, Senior Area Chair, and Program Chairs,

We sincerely appreciate the time and effort you dedicate to meta-reviewing our paper. We understand that the current situation significantly increases your workload. To assist you as much as we can, we provide a summary of our author-reviewer discussion. During the discussion period, we **thoroughly addressed all reviewers' concerns** and conducted **extensive new quantitative experiments** .

**On 11.23 AoE(Before 11.26 AoE , to our knowledge, the time of the OpenReview information leak)**， the ratings are **Uniformly Positive: 6, 6, 6** (Reviewer QFw6 raised their score from 4 to 6).

1. **Reviewer QFw6 (Score: 4 $\to$ 6) [Confidence 5]**

**Statue: Following our detailed clarification and additional experiments, the reviewer explicitly acknowledged that most of their concerns were resolved and raised their score from 4 to 6.**

The reviewer initially questioned the novelty compared to SEED-X, the architectural classification, and potential data contamination. We addressed these through rigorous discussion and new experiments:

- **Addressed "Benchmark Hacking" Concern:** We performed a rigorous `Data Exclusion Experiment`. We retrained ORION **excluding the controversial BLIP-3o dataset**. The model maintained a GenEval score of **0.81** (comparable to the original 0.82), proving our performance stems from architectural strengths, not data memorization.
- **Expanded Evaluation Scope:** To address the concern of relying solely on GenEval, we `evaluated on non-semantic benchmarks`. ORION achieved competitive results on **WISE** (World Knowledge, **0.52**) and **MJHQ-FID** (Image Quality, **5.58**), demonstrating robust general capabilities.
- **Clarified Architecture & Novelty:** We provided a `detailed architectural analysis` distinguishing ORION’s **native token regression** from SEED-X’s fixed queries, and clarified that the "semantic-structural conflict" we solve occurs specifically at the **shared last hidden state**.

2. **Reviewer FMab (Score: 6) [Confidence 4]**

**Status: We addressed all raised concerns by conducting the requested quantitative evaluations and ablation studies, though the discussion period concluded before we could receive a follow-up.**

This reviewer was positive but requested quantitative evidence for image editing and a deeper ablation study.

- **New Quantitative Editing Benchmark:** We trained a `specialized Diffusion Decoder` during the rebuttal to enable quantitative evaluation. We evaluated on 1,000 held-out samples, where ORION achieved **CLIP-T 0.319** and **L1 Distance 0.137**, outperforming or matching diffusion-based baselines like InstructPix2Pix and OmniGen.
- **Factorial Ablation Study:** We conducted a `new ablation` to isolate "Decoupling" vs. "Alignment." The results (Linear: 71.5 $\to$ Linear+Align: 77.8 $\to$ MLP+Align: **83.7**) quantitatively proved that while alignment is beneficial, **non-linear decoupling is essential** to fully resolve the representation bottleneck.
- **Comparative Analysis:** We added `discussions on Emu3 and GLaMM`, highlighting ORION's advantage in using continuous token regression over discrete quantization for preserving understanding.

3. **Reviewer YV3K (Score: 6) [Confidence 4]**

**Status: We provided all requested technical details and verified the model's emergent capabilities, though the deadline for reviewer replies passed before we received a follow-up.**

The reviewer asked for technical specifics on the consistency loss and evidence of "emergent" capabilities.

- **Verified Emergent Capabilities:** To prove the model's generalization, we designed a `Cross-Lingual Probe`. We translated GenEval prompts into **Chinese** (a language not used for generation training). ORION achieved a score of **0.75** (vs. English 0.82), quantitatively validating the transfer of semantic knowledge from the understanding module to generation.
- **Sensitivity Analysis:** We provided a `parameter sweep` for the KL loss weight ($\lambda_{KL}$), identifying **0.01** as the optimal balance point between understanding and generation, and clarified all Teacher Model details (frozen Qwen2.5-VL, Temp=1.0).
- **Data Governance:** We confirmed strict `data decontamination protocols` and licensing compliance for all datasets (including FLUX regenerated images).

We are encouraged that the reviewers recognized the value of our work, citing it as **"conceptually simple yet targeted"** (Reviewer YV3K), noting that it **"clearly formalizes the semantic-structural conflict"** (Reviewer FMab, YV3K) and recognition of **our solid performance on benchmarks(Reviewer FMab, QFw6)**. With the additional experiments confirming robustness, data integrity, and editing fidelity, we believe ORION represents a solid contribution to unified MLLM architectures.

We remain available for any further questions. Thank you for your supervision.

Best regards,

The Authors

---

### Meta-Review · Area_Chair_X1GL · 2026-01-07

**Summary:**

The reviews are uniformly positive. There are concerns around benchmark, novelty, and other minor concerns. The authors did a great job addressing them during rebuttal.

**Reviewer Concerns:**

Reviewer QFw6. All major concerns were addressed in the rebuttal. Concerns regarding novelty vs. SEED-X, architectural clarity, and potential benchmark hacking/data contamination were resolved through new architectural analysis, a data-exclusion experiment (GenEval 0.81 vs. 0.82 without BLIP-3o), and expanded evaluations on WISE and MJHQ-FID. The reviewer explicitly acknowledged resolution of most issues and increased their score from 4 to 6. No outstanding concerns remain.

Reviewer FMab. The reviewer’s requests for quantitative image-editing evaluation and deeper ablations were fully addressed via a new diffusion-decoder benchmark, factorial ablation isolating decoupling vs. alignment, and additional comparative discussion. No technical concerns remain.

Reviewer YV3K. Requests for technical details and evidence of emergent capabilities were addressed through a cross-lingual GenEval probe, KL-loss sensitivity analysis, and clarification of teacher and data-governance details. No outstanding concerns remain.

**Reviewer Scores:**

Reviewer QFw6 increase 4->6

---

### Decision · Program_Chairs · 2026-01-26

Accept (Poster)